# Understanding crystallization pathways leading to manganese oxide polymorph formation

Bor-Rong Chen[1], Wenhao Sun[2,3], Daniil A. Kitchaev [4], John S. Mangum[5], Vivek Thampy[1], Lauren M. Garten[6], David S. Ginley[6], Brian P. Gorman[5], Kevin H. Stone [1], Gerbrand Ceder[2,3], Michael F. Toney [1,7] & Laura T. Schelhas [7]

Hydrothermal synthesis is challenging in metal oxide systems with diverse polymorphism, as reaction products are often sensitive to subtle variations in synthesis parameters. This sensitivity is rooted in the non-equilibrium nature of low-temperature crystallization, where competition between different metastable phases can lead to complex multistage crystallization pathways. Here, we propose an ab initio framework to predict how particle size and solution composition influence polymorph stability during nucleation and growth. We validate this framework using in situ X-ray scattering, by monitoring how the hydrothermal synthesis of $MnO_2$ proceeds through different crystallization pathways under varying solution potassium ion concentrations ($[K^+] = 0$, 0.2, and 0.33 M). We find that our computed size-dependent phase diagrams qualitatively capture which metastable polymorphs appear, the order of their appearance, and their relative lifetimes. Our combined computational and experimental approach offers a rational and systematic paradigm for the aqueous synthesis of target metal oxides.

[1] Stanford Synchrotron Light Source, SLAC National Accelerator Laboratory, Menlo Park, CA 94025, USA. [2] Materials Science Division, Lawrence Berkeley National Laboratory, Berkeley, CA 94720, USA. [3] Department of Materials Science and Engineering, UC Berkeley, Berkeley, CA 94720, USA. [4] Department of Materials Science and Engineering, Massachusetts Institute of Technology, Cambridge, MA 02139, USA. [5] Metallurgical and Materials Engineering, Colorado School of Mines, Golden, CO 80401, USA. [6] National Renewable Energy Laboratory, Golden, CO 80401, USA. [7] Applied Energy Programs, SLAC National Accelerator Laboratory, Menlo Park, CA 94025, USA. These authors contributed equally: Bor-Rong Chen, Wenhao Sun. Correspondence and requests for materials should be addressed to G.C. (email: gceder@berkeley.edu) or to M.F.T. (email: mftoney@slac.stanford.edu) or to L.T.S. (email: schelhas@slac.stanford.edu)

Rapid improvements in the theory[1,2] and characterization[3,4] of inorganic compounds have led to tremendous advances in our understanding of structure-property relationships. However, similar quantitative and predictive theories for understanding synthesis–structure relationships remain largely undeveloped. In particular, finding the conditions required to synthesize a particular stable or metastable phase can be a time consuming trial-and-error process. While phase diagrams offer guiding insights on the relative stability of reaction end-products, little is known about the energy landscape that a material traverses as it nucleates and grows. There is growing evidence of the complexity of this energy landscape, as well-crystallized transient metastable phases are often observed to form before the equilibrium phase, during both solid-state[5–8] and aqueous synthesis[9,10]. Furthermore, minor variations in synthesis conditions can affect the lifetimes of these metastable phases[11], or even redirect a crystallization pathway through very different metastable phases[12,13]. Understanding how synthesis parameters influence the formation and persistence of these metastable intermediates offers a roadmap towards rational materials synthesis, as metastable phases can often exhibit superior properties to their stable counterparts[14–17].

Recent experimental and theoretical studies indicate that metal oxides can undergo dramatic thermodynamic crossovers in polymorph stability, as a function of both particle size and composition. Small particles exhibit large surface-area-to-volume-ratios, meaning a bulk metastable phase with low surface energy can become thermodynamically-stabilized at the nanoscale[18–20]. Alkali ions from solution can preferentially intercalate into specific metal-oxygen frameworks, stabilizing a polymorphic structure at off-stoichiometric compositions[21]. These two properties of size and composition are especially relevant during hydrothermal synthesis; as all materials nucleate and grow through a nanoscale size regime, and the intercalation of impurity ions from solution can stabilize off-stoichiometric polymorphs during materials growth. Here, we construct aqueous phase stability diagrams that explicitly incorporate the thermodynamic effects of particle size and composition[22]. These size- and composition-dependent Pourbaix diagrams offer a general theoretical framework to rationalize which non-equilibrium phases appear during the aqueous precipitation of transition metal oxides, and the order that they appear; laying an important foundation for a predictive theory of materials synthesis.

To validate our theoretical framework, we experimentally conduct in situ hydrothermal synthesis of manganese oxide ($MnO_2$) at varying conditions, and compare experimentally observed crystallization pathways to the ab initio computed phase diagrams. We choose $MnO_2$ because it exhibits diverse polymorphism[23,24] and many of the structural polymorphs are of interest for specific applications, such as Li-ion battery cathodes, catalysts, capacitors, and molecular sieves, etc[24–27]. Empirical recipes for the synthesis of specific manganese oxide polymorphs have been cataloged[27–29], and generally exhibit two common themes; they involve quenching a multistage crystallization at intermediate times to retrieve a metastable phase[30], and often require a structure-directing impurity ion to drive the formation of a specific phase[21]. However, it is difficult to extend these recipes to other metal oxide systems, as these empirical recipes have not been rationalized from general thermochemical principles. Here, we characterize in situ the hydrothermal synthesis of manganese oxides under varying $[K^+]$ in solution, to direct the hydrothermal reduction of a $MnO_4^-$ based precursor down different crystallization pathways. In situ X-ray wide-angle scattering (WAXS) allows us to monitor the evolution of crystalline intermediates at different $[K^+]$, and compare the observed crystallization pathways against ab initio computed phase diagrams with axes of particle size and solution $[K^+]$.

Under all conditions, crystallization pathways proceed through intermediate metastable phases. We find that the order of observed metastable $MnO_2$ phases corresponds qualitatively with a growth progression through the computed size-dependent phase diagrams, at the respective $[K^+]$. This suggests that the observed metastable phases nucleate under instantaneous local thermodynamic conditions where they were once stable, and then grow into conditions where they become metastable, consistent with a concept of remnant metastability[31]. Our combined experimental and theoretical study emphasizes the importance of particle size and solution chemistry in influencing the free-energy landscape of non-equilibrium crystallization, which directly governs polymorph-selection during hydrothermal synthesis. In addition, it offers evidence that a concept of remnant metastability can serve as a guiding principle to design rational synthesis pathways; prompting a more deliberate search for relevant thermodynamic parameters that govern structure-selection during materials synthesis.

## Results

**Theory of crystallization pathways.** Crystallization initiates from a high-energy precursor, and traverses through a metastable free-energy landscape in the multistage nucleation and growth of a bulk macroscopic crystal[32]. Mapping this free-energy landscape lays a conceptual foundation for understanding structure-selection during multistage crystallization. Here, we discuss how surface energies and solution chemistry influence this free-energy landscape, by stabilizing specific $MnO_2$ phases at various sizes and off-stoichiometric compositions.

Before reaching the bulk equilibrium state, all materials first nucleate and grow through the nanoscale regime, where surface energy contributions are significant. Small particle sizes, or large $1/R$ ($R$ is the radius), can stabilize a bulk-metastable polymorph with low surface energy, leading to preferential nucleation of this metastable phase over the equilibrium phase. Figure 1 shows the size-dependent free-energy ($\Phi$) versus the particle size ($1/R$), illustrating how the interplay between bulk and surface energies can lead to multistage crystallization. Starting from a supersaturated solution (gray line), a metastable phase with low surface energy, $M$, is the lowest free-energy phase at small sizes, meaning that $M$ requires the smallest size-fluctuation to nucleate. After its nucleation, $M$ reduces its free energy by crystal growth, consuming metal solute ions in the process. Because the barrier to crystal growth is smaller than the barrier to nucleation of a new phase, $M$ may grow to a size region where the $M$ is no longer size-stabilized. At this point, there is a thermodynamic driving force for a phase transformation from $M$ onto $S$, the lower free-energy phase. The induction time to nucleate $S$ depends on the surface energy of $S$ and the thermodynamic driving force for transformation, $\Delta\Phi_{M->S}$[11]. A smaller $\Delta\Phi$ leads to longer induction times, which would prolong the persistence of the metastable phase $M$. Once $S$ nucleates, the entire system can reduce its free energy by the dissolution of $M$, and subsequent reprecipitation onto $S$.

In the manganese oxides, we previously showed that the bulk free-energy of each polymorph, $\Phi$, can also be influenced by alkali ion intercalation, which stabilizes different polymorphic frameworks at off-stoichiometric compositions[21,23]. For example, solution $K^+$ ions can intercalate into the $\alpha$- (tunnel) and $\delta$- (layered) $MnO_2$ polymorphs, stabilizing them as a function of aqueous $K^+$ activity; and other $MnO_2$ polymorphs can be stabilized by $Li^+$, $Na^+$, $Ca^{2+}$, etc[21]. Intercalation of ions into polymorphic frameworks occurs ubiquitously in transition metal

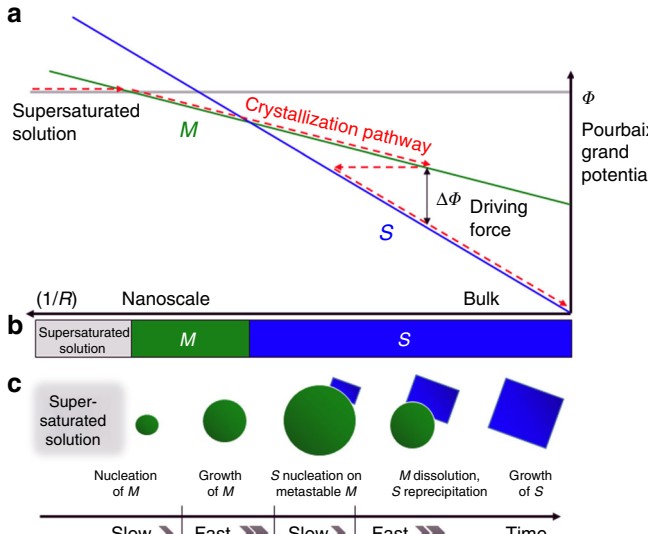

**Fig. 1** Schematic representation of remnant metastability in a crystallization pathway. **a** Free-energy of three phases (supersaturated solution (gray), $M$ (green), $S$ (blue)) as a function of the surface-area-to-volume ratio, $1/R$ ($R$ is a particle radius). The gray line corresponds to the free-energy of a supersaturated solution, green is a metastable phase $M$ that is size-stabilized by a low surface energy (given by the slope), and blue is the bulk equilibrium phase $S$, with high surface energy. **b** Phase diagram in the $1/R$ axis created from the projection of lowest free-energy phases. **c** A multistage crystallization pathway (red arrow in **a**) proceeds downhill in energy, but phase transformations are limited by nucleation. Crystal growth of $M$ prior to the induction of $S$ means $M$ can grow into a size-regime where phase $M$ is metastable. $S$ will then nucleate, and quickly grow by consuming $M$ via dissolution-reprecipitation. The characteristic length scale of size-driven phase transitions lies in the 2 nm–50 nm range

oxide systems, meaning solution chemistry can be considered a general handle for tuning bulk polymorph stability. To account for both the size and composition of manganese oxide particles, we can define the free-energy of a manganese oxide particle in water by a full size-dependent Pourbaix grand potential[22], $\Phi$, which is a function of the pH; the redox potential, $E$; the potassium ion activity which can be approximated by the potassium ion concentration, $[K^+]$; and the particle size, signified by the surface-area-to-volume-ratio, or $1/R$, where $R$ is the particle radius. The complete equation for $\Phi$ is provided in the Methods. The thermochemical data for the K-Mn-O-H system were obtained from a comprehensive set of density functional theory calculations of pristine and off-stoichiometric bulk and surface energies. Our DFT calculations use the recently-developed SCAN metaGGA functional[33], which is necessary to accurately reproduce energy differences between $MnO_2$ polymorphs.

**Role of $[K^+]$ in the crystallization pathways**. We perform a joint computational and experimental study to quantify the thermodynamic role of $[K^+]$ as a structure-directing agent in the multistage crystallization of manganese oxide polymorphs to examine the robustness of the theoretical framework. Experimentally, we perform in situ WAXS to probe the hydrothermal reduction of a $MnO_4^-$ precursor at three potassium concentrations: $[K^+] = 0$, 0.2, and 0.33 M, denoted as $P_{K=0}$, $P_{K=0.2}$, and $P_{K=0.33}$, respectively. Crystallization reactions were performed between pH 1–2. We measured the redox potential of these aqueous solutions to be $+1.2$ V vs the standard hydrogen electrode (SHE), which decreased to $+0.5$ V after the reaction. The pH values and the

change in redox potential are discussed in Supplementary Table 1.

We compute that there are four relevant $MnO_2$ phases at these solution conditions; $\beta$-$MnO_2$ (pyrolusite), $\alpha$-$MnO_2$ (hollandite), R-$MnO_2$ (ramsdellite), and the layered $\delta$-$MnO_2$ (birnessite/vernadite). We note that the layered $\delta$-phase encompasses a large family of structures with varying hydration, ion intercalation, and layer spacings[34,35]. Throughout this work, we distinguish two of the experimentally-resolved layered structures in this family as $\delta'$ and $\delta''$. Another family of phases is the so-called intergrowth phases, which is a disordered intergrowth between R-$MnO_2$ and $\beta$-$MnO_2$ on a shared oxygen sub-lattice. In this paper, we name the observed intergrowth phase $\gamma'$ to distinguish this $\gamma'$ phase from the idealized $\gamma$-$MnO_2$ structure, which is an ordered 50/50 intergrowth between R-$MnO_2$ and $\beta$-$MnO_2$[21]. Detailed information of all the $MnO_2$ phases relevant in this study is provided in Supplementary Table 2, and the structure files can be found in Supplementary Data 1–5.

To rationalize the influence of $[K^+]$ on $MnO_2$ crystallization, we calculate bulk and size-dependent phase diagrams at varying $K^+$ chemical potential. The bulk phase diagram at $E = 1.2$ V (Fig. 2(a)) indicates that $\beta$-$MnO_2$ is the equilibrium phase at low $[K^+]$, and that $\alpha$-$K_xMnO_2$ can be stabilized against $\beta$-$MnO_2$ under high $[K^+]$. The size-dependent phase diagram (Fig. 2(b)), taken at $E = 1.2$ V and fixed pH = 2, reveals a more complicated phase stability landscape, showing that the low surface energies of $\delta$-$MnO_2$ and R-$MnO_2$ can stabilize these bulk-metastable phases at small sizes and low $[K^+]$, and that a $\delta$-$K_{0.33}MnO_2 \cdot 0.66H_2O$ phase can be size-stabilized at higher $[K^+]$. Cutaways of $\Phi$ as a function of particle size, analogous to Fig. 1, are shown in Fig. 2(c) (low $[K^+]$), and 2(d) (high $[K^+]$). While bulk energetics would only show the stability of $\beta$-$MnO_2$ or $\alpha$-$K_xMnO_2$ at these conditions, the size-dependent Pourbaix potential reveals the metastable energy landscape the K-Mn-$H_2O$ system traverses as the particles nucleate and grow. Taken together, these diagrams map the free-energy landscape of $MnO_2$ crystallization under the influence of varying $[K^+]$ and particle size, and provide a foundation towards rationalizing the order and lifetime of polymorphs in the phase evolution of the reaction pathways.

**Crystallization pathway in low $[K^+]$ region**. The in situ WAXS results for $P_{K=0}$ are summarized in Fig. 3. The evolution of intermediates along the reaction pathways are illustrated by the partially background-subtracted X-ray scattering profiles shown in Fig. 3(a), and the evolution of the phase fractions shown in Fig. 3(b); details of background subtraction and phase fraction calculations can be found in Supplementary Note 3 and 4. We observe a three-stage crystallization along $P_{K=0}$, in which $\delta''$ (belonging to the $\delta$-family), $\gamma'$ (belonging to the intergrowth family), and $\beta$-$MnO_2$ appear in a successive order that qualitatively corresponds to growth through the size-dependent phase diagram in Fig. 2(b).

The first phase we observe is characterized by two diffraction peaks at $q = 2.58$ and $4.45$ Å$^{-1}$. The position and asymmetrical shape of the peaks are consistent with the intra-layer diffraction peaks of the layered $\delta'$-$MnO_2$ phase observed in $P_{K=0.2}$ and $P_{K=0.33}$ (Fig. 4(g)) and discussed in detail in the following section. However, the $\delta''$-phase is missing the two interlayer diffraction peaks present in the $\delta'$ phase. This diffraction pattern suggests that, while the $\delta''$-phase is structurally related to the $\delta'$-phase, the former may lack the long-ranged interlayer order as in the latter. Most likely, the $\delta''$-phase consists of small platelet structures, due to the absence of a highly asymmetric peak[36] at high q that results from isolated $MnO_2$ layers. This disorder may

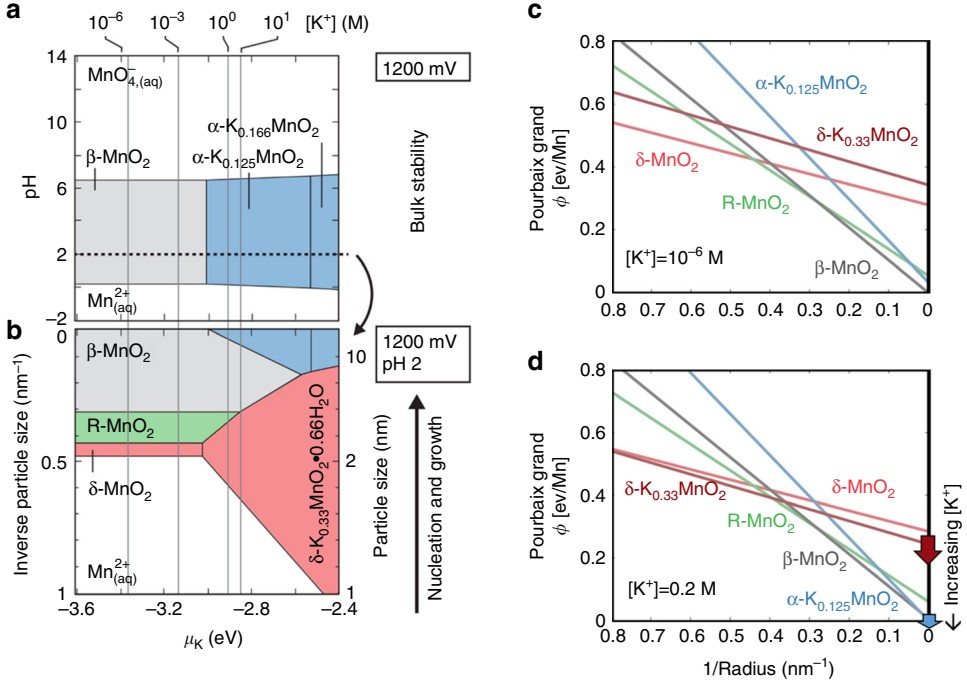

**Fig. 2** Phase diagrams of K-Mn oxide systems at $E = 1.2$ V. **a** pH-$\mu_K$ slice at $1/R = 0$ indicating bulk stabilized phases. **b** $1/R$-$\mu_K$ slice at pH = 2 illustrating the size-stabilized phases. **c**, **d** Pourbaix grand free-energies versus $1/R$, showing the metastable energy landscape of crystal growth where $[K^+] = 10^{-6}$ M (**c**) and $[K^+] = 0.2$ M (**d**). **d** is representative of pathways at both $[K^+] = 0.2$ M and $[K^+] = 0.33$ M due to their similar $[K^+]$. The three reaction conditions are referred to as $P_{K=0}$, $P_{K=0.2}$, and $P_{K=0.33}$, respectively, based on their $[K^+]$. $d(\Phi_{\delta-K_{0.33}MnO_2} - \Phi_{\alpha-K_{0.125}MnO_2})/d\mu_K = -0.21$, which means that the driving force for the transformation from the δ to the α phase decreases at a larger $[K^+]$. The red and blue arrows indicate the trend that the slope of $\Phi_{\delta-K_{0.33}MnO_2}$ and $\Phi_{\alpha-K_{0.125}MnO_2}$ changes with increasing $[K^+]$

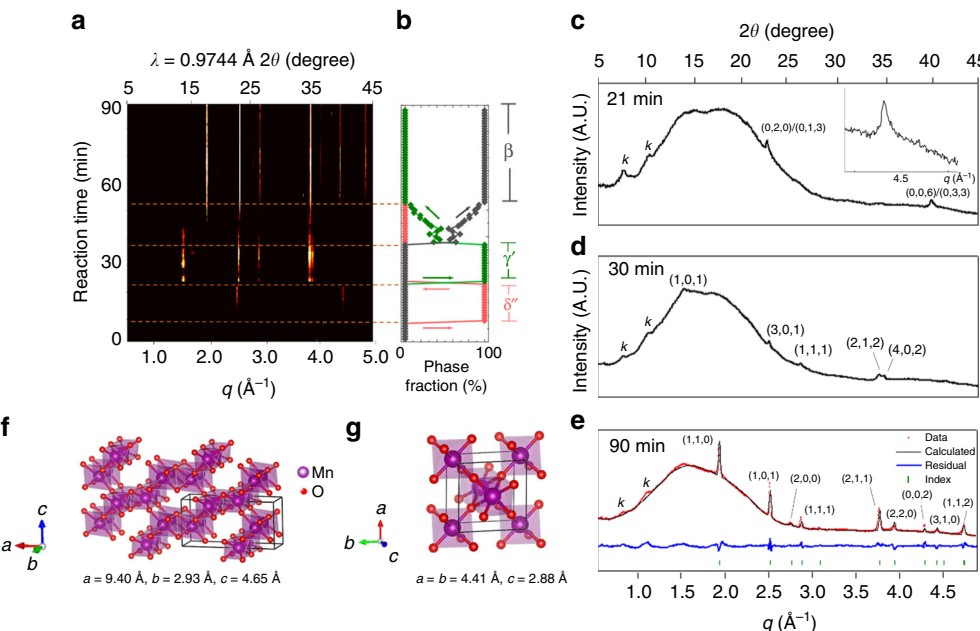

**Fig. 3** In situ X-ray scattering analysis of $P_{K=0}$. **a** The evolution of X-ray scattering profiles with time along $P_{K=0}$. $q$ is related to the diffraction angle ($\theta$) and incident wavelength ($\lambda$) by $q = \frac{4\pi}{\lambda}\sin\theta$ and $\lambda = 0.9744$ Å. **b** Phase fraction evolution calculated from the intensities of diffraction peaks in **a**. **c**-**e** X-ray scattering profiles of the δ'' (**c**), γ' (**d**), and β (**e**) phases, respectively. The inset figure in **c** shows the asymmetrical tail of the peak at $q = 4.45$ Å$^{-1}$. The weak peaks around $q = 0.9$ and 1.2 are caused by a kapton film covering the capillary reactor (marked by $k$ in **c**-**e**). The broad peak around $q = 1.5$-2.5 Å$^{-1}$ is caused by imperfect background subtraction (see Supplementary Notes 3 for details). **f** Structure of a pristine R phase. The γ' phase discussed in this paper is an R phase with a random amount of β (1 × 1 tunnel) intergrowth. **g** Refined structure of the β phase (see Supplementary Notes 5 for refinement details)

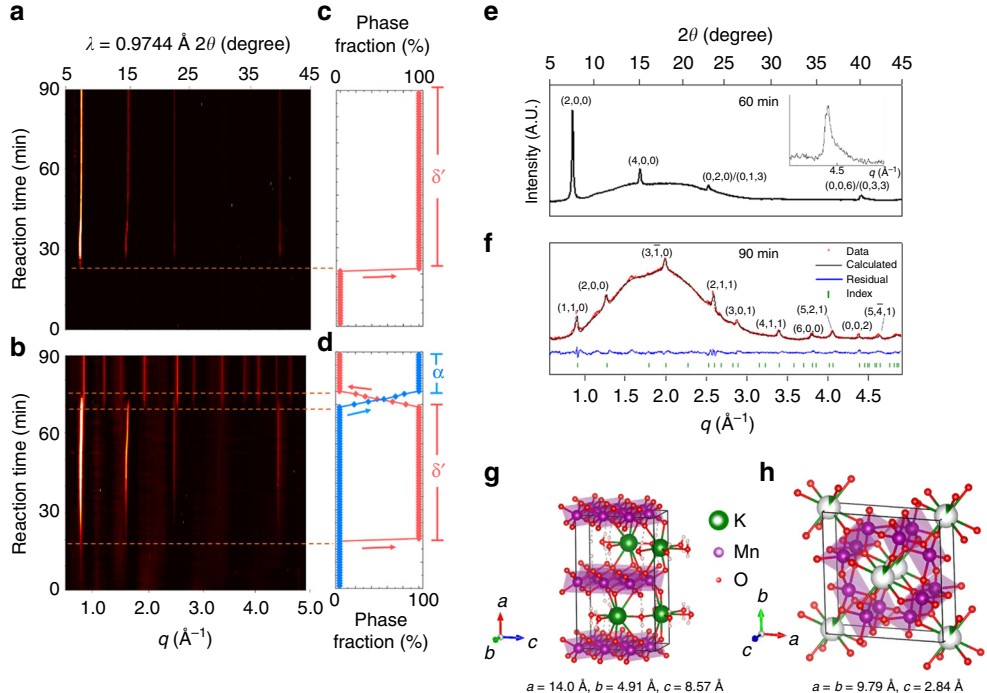

**Fig. 4** In situ X-ray scattering analysis of $P_{K=0.33}$ and $P_{K=0.2}$. **a, b** The evolution of X-ray scattering profile with time during hydrothermal synthesis along $P_{K=0.33}$ (**a**) and $P_{K=0.2}$ (**b**). **c, d** Phase fraction evolution calculated from the intensities of diffraction peaks in **a**, **b**, respectively. **e, f** X-ray scattering profiles of the δ′ phase (**e**) and α phase (**f**). The inset figure in **d** shows the asymmetrical tail of the peak at $q = 4.45$ Å$^{-1}$. The broad peak around $q = 1.5$–$2.5$ Å$^{-1}$ is caused by imperfect background removal, and is discussed in Supplementary Note 3. **g, h** Refined structures of the δ′ phase (**g**) and α phase (**h**). Manganese (purple), potassium (green), and oxygen (red) are represented by spheres, and the unit cell is identified by gray lines (see Supplementary Notes 5 for refinement details). The partially filled spheres represent the site fraction of the atoms

be caused by the lack of K$^+$ in the system, which promotes strong stacking ordering due to the strong ionic K-O bonds[35,37].

The second intermediate phase, with the diffraction profile shown in Fig. 3(d), is a ramsdellite (R)-like framework $(1 \times 2)$ tunnel (Fig. 3(f)) with some amount of random pyrolusite $(1 \times 1)$ tunnel intergrowths. We identify this phase as a ramsdellite structure containing de Wolff defects and microtwinning[38,39], and refer to it as γ′-MnO$_2$. The fraction of $(1 \times 1)$ intergrowth in the γ′-MnO$_2$ that we observed was approximately 30%, and the detailed calculation can be found in Supplementary Note 6.

The final phase to appear has the scattering profile shown in Fig. 3(e) and is refined to be the β-MnO$_2$ phase in Fig. 3(g), consistent with previous reports in low [K$^+$] synthesis conditions[40,41]. This trend of transformation from δ″ to γ′ and then to β-MnO$_2$ also agrees with previous reports of Na-incorporated birnessite gradually transforming to $(1 \times 2)$ tunnel, and finally a $(1 \times 1)$ tunnel or β-MnO$_2$ structure[40].

The observed order of MnO$_2$ phases from δ″ → γ′ → β can be qualitatively mapped onto a growth progression through the size-dependent phase diagram of Fig. 2(b) at [K$^+$] < 10$^{-6}$ M. δ-MnO$_2$ forms first, because δ-MnO$_2$ is stabilized at small sizes by its low surface energy. The R-, γ′- and β- phases of MnO$_2$ all share the same oxygen sub-lattice, and differ from one another by the manganese occupation of the octahedral lattice sites. The γ′-family represents a disordered structural continuum between the $(1 \times 2)$ tunnel R-MnO$_2$ and the $(1 \times 1)$ tunnel β-MnO$_2$, and in our experiment the observed γ′-MnO$_2$ has 70% R-MnO$_2$ character. Because the observed γ′-MnO$_2$ is disordered, its thermodynamic properties cannot be directly computed in DFT, but the γ′-MnO$_2$ bulk and surface energies are likely reasonably approximated by the structurally-similar, ordered R-MnO$_2$ phase. In the size-dependent phase diagram, using the

ordered R-MnO$_2$ as a proxy for the experimentally observed disordered γ′ phase qualitatively captures the thermodynamic driving force in a size-driven phase progression from δ″ → γ′. Finally, because β-MnO$_2$ is the lowest bulk energy phase, the disordered γ′ intermediate eventually transforms into the topotactically-related β-phase. This is a transformation pathway similar to previous experimental results[42]. Therefore, our experimentally observed phase progression qualitatively reflects a remnant metastability interpretation of the size-dependent phase diagram, where the major structural motifs of the experimentally observed δ″, γ′, and β phases are captured by ordered δ-, R-, and β- phases computed by DFT. The consistency between experiment and the size-dependent phase diagram supports the utility of remnant metastability as a guiding principle for the synthesis planning of a multistage crystallization.

**Crystallization pathways in the higher [K$^+$] region.** Aqueous K$^+$ ions can be important structure-directing agents in the precipitation of MnO$_2$ phases, as intercalation of potassium into tunnel frameworks changes the equilibrium phase from a $(1 \times 1)$ β-MnO$_2$ phase to $(2 \times 2)$ α-MnO$_2$ phase[21,41,43], and also increases the stability of δ′-MnO$_2$ by K$^+$-intercalation between the layers. We will demonstrate that an intermediate [K$^+$] indeed promotes the crystallization of the α- phase, but if the [K$^+$] is too high, it can increase the persistence of the metastable δ- phase, which illustrates the competition between thermodynamic stability and the kinetics of phase transformations.

To validate the influence of K$^+$ on redirecting the crystallization pathway, we perform in situ hydrothermal synthesis with [K$^+$] = 0.2 M and [K$^+$] = 0.33 M. Figure 4(a–d) show the in situ WAXS data and phase evolution versus time for the first 90 min

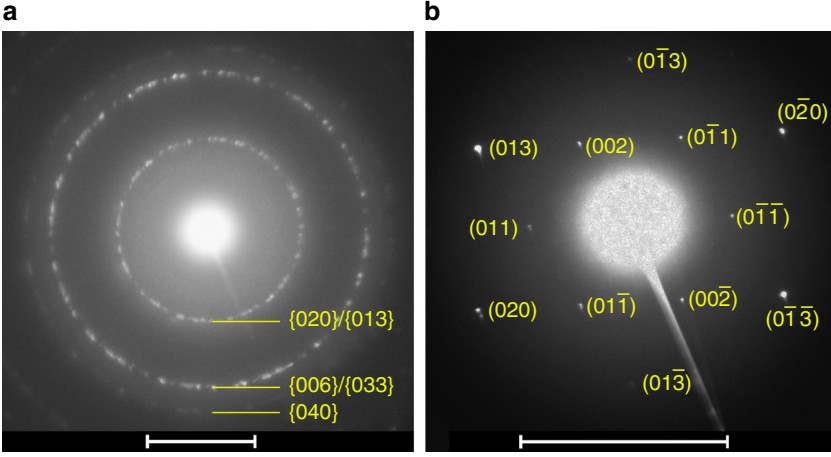

**Fig. 5** Quenched phases. Selected area electron diffraction (SAED) patterns of the quenched δ′ (**a**) and δ′′ (**b**) phases quenched from $P_{K=0.2}$ and $P_{K=0}$, respectively. The scale bars are 5 nm$^{-1}$ in both of the figures

of the reactions. $P_{K=0.2}$ proceeds via two crystallization stages over the course of 90 min; while for $P_{K=0.33}$, we observe only the first stage, which does not transform to the second stage even after a 70 h reaction (see Supplementary Figure 2).

We resolve the initial phase to be in the same layered δ-MnO$_2$ phase family as found in $P_{K=0}$. This family of compounds is represented by the general chemical formula K$_x$MnO$_2$ · yH$_2$O ($x$ = 0.25–0.75), where the amount of K$^+$ intercalation and hydration changes the interlayer spacing of the MnO$_2$ sheets[44]. To refine the structure and composition of the observed δ-phase, we first determine the d-spacing and relative intensities of the diffraction peaks from the WAXS data. As extracted from the diffraction profile in Fig. 4(e), the strongest diffraction peak, at $q$ = 0.86 Å$^{-1}$, corresponds to an interlayer d-spacing of 7.3 Å. We next use DFT to resolve which K$_x$MnO$_2$·yH$_2$O stoichiometry gives the best agreement with this diffraction profile, predicting a final δ′ stoichiometry of K$_{0.33}$MnO$_2$-0.66H$_2$O, which is a P2-type layered structure, as shown in Fig. 4(g). Inductively coupled plasma (ICP) measurements on products synthesized ex situ were performed and confirmed the refined stoichiometry (see Supplementary Note 7 and 8 for details).

To confirm the structure of the δ′ phase, we match the plane spacings with the positions of the experimental diffraction profile in Fig. 4(e). The scattering profile of δ′ consists of four major peaks. The two peaks at $q$ = 0.86 Å$^{-1}$ (2,0,0), and 1.72 Å$^{-1}$ (4,0,0) correspond to the interlayer spacing, and the two peaks at 2.58 Å$^{-1}$ ((0,2,0)/(0,1,3)), and 4.45 Å$^{-1}$ ((0,0,6)/(0,3,3)) correspond to intra-layer spacing of hexagonally close-packed Mn atoms in individual MnO$_2$ layers in the theoretically-predicted K$_{0.33}$MnO$_2$-0.66H$_2$O structure. For detailed $hkl$ indexing of the δ′ phase, see Supplementary Figure 5. The two peaks at $q$ = 2.58 and 4.45 Å$^{-1}$ show asymmetric tails (see the insert Fig. 4(e)), suggesting the presence of stacking faults between the MnO$_2$ layers[45–47]. However, the stacking faults and the limited number of diffraction features in the δ′ phase preclude reliable structure determination with Rietveld refinement.

In determining the end product for $P_{K=0.2}$ (Fig. 4(f)), we refine the scattering profile to α- MnO$_2$. The refined chemical formula of the α phase is K$_{0.10 \pm 0.02}$MnO$_2$, which matches the stoichiometry of the computationally predicted α-K$_{0.125}$ MnO$_2$ phase, shown in Fig. 4(h). ICP confirms that the α phase synthesized ex situ has a K to Mn ratio of $0.10 \pm 0.01$, consistent with the occupancy obtained from the Rietveld refinement (see Supplementary Note 5 and 8 for the details in refinements and ICP measurements).

On the computed size-dependent phase diagram, the stable phases as a function of increasing size at [K$^+$] ~10$^{-1}$ M proceeds as δ′-K$_{0.33}$MnO$_2$ → R-MnO$_2$ → β-MnO$_2$ → α-K$_{0.125}$MnO$_2$. The in situ observation of an intermediate δ-phase, followed by an α-phase, is consistent with the order progression as seen on the size-dependent phase diagram. However, why does neither the R-MnO$_2$ nor the β-MnO$_2$ phases form? At [K$^+$] ~10$^{-1}$ M, both R-MnO$_2$ and β-MnO$_2$ are size-stabilized between 2 and 10 nm, as seen in the size-dependent phase diagram in Fig. 2(b). The crystallites in the observed δ-K$_{0.33}$MnO$_2$ phase are estimated to be at least 30 nm in diameter by Scherrer analysis; much larger than the R-MnO$_2$ and β-MnO$_2$ stability regimes. Because the energy barrier to nucleation is larger than the barrier to crystal growth, the δ-phase likely nucleated and grew beyond the size-stability regime of R- and β-MnO$_2$, so that these intermediate size-stabilized phases were skipped during multistage crystallization. Similar phenomena occur in other metal oxide systems, for example, in TiO$_2$, the brookite polymorph is size-stabilized between anatase and rutile[48], while in Fe$_2$O$_3$, the ε-Fe$_2$O$_3$ phase is size stabilized between γ-Fe$_2$O$_3$ and α-Fe$_2$O$_3$[49]. Although experimentally, the formation of brookite TiO$_2$ and ε-Fe$_2$O$_3$ during aqueous precipitation is rare, both brookite TiO$_2$ and ε-Fe$_2$O$_3$ can be stabilized by annealing at constrained size regimes[50,51]. Hence, if crystal growth in our MnO$_2$ system can be inhibited, it is plausible that a phase evolution of δ → R → β → α can be observed here as well.

From thermodynamic considerations, the δ′-phase is clearly metastable with respect to the α- phase, and increasing [K$^+$] should only further increase the stability of the α phase. However, our experimental results demonstrate that the δ′ to α transformation occurs in only 20 min in the lower [K$^+$] pathway ($P_{K=0.2}$) but that the δ′-phase can persist for more than 70 h in the higher [K$^+$] pathway ($P_{K=0.33}$). While a higher aqueous [K$^+$] lowers the free-energy of both α-K$_{0.125}$MnO$_2$ and δ′-K$_{0.33}$MnO$_2$, the free-energy of the potassium-rich δ′-K$_{0.33}$MnO$_2$ phase is lowered more than that of the α-K$_{0.125}$MnO$_2$. This indicates that higher [K$^+$] reduce the driving force ($\Delta\Phi$) required for the δ′-K$_{0.33}$MnO$_2$ to α-K$_{0.125}$MnO$_2$ transformation (blue/red arrows Fig. 2(d)). The result is that even though a high K$^+$ activity should ultimately make α-K$_{0.125}$MnO$_2$ more stable, it concurrently increases the stability of the metastable δ′-K$_{0.33}$MnO$_2$ intermediate, inhibiting the kinetics of the δ′-to-α transformation, and prolonging the persistence of the δ′-K$_{0.33}$MnO$_2$[11]. Our observations illustrate the competing effects of bulk thermodynamics and transformation kinetics in the progression of a multistage crystallization pathway,

and highlights the importance of determining free-energy differences between reactants, metastable intermediates, and equilibrium end-products when understanding multistage crystallization.

**Synthesizing desired intermediate phases**. We have so far shown that it is possible to predict which metastable phases appear during crystallization. Next, to demonstrate if these metastable phases are obtainable, we isolate desired metastable phases by quenching the solution during a crystallization reaction. We quench $P_{K=0}$ and $P_{K=0.2}$ by stopping the reaction by breaking the glass capillary in half at appropriate reaction times to stop the reaction. The time points for harvesting the $\delta'$ and $\delta''$ phases are determined by the progression of intermediate phases observed in the in situ scattering experiments, as shown in Fig. 4(b) and Fig. 3(a).

Using transmission electron microscopy (TEM), we confirm that we successfully isolated the transient metastable phases $\delta'$ and $\delta''$. Figure 5 shows the selected area electron diffraction (SAED) patterns of the quenched $\delta'$ and $\delta''$ phases. The reflections in Fig. 5(a) located on the diffraction rings labeled {020}/{013} and {006}/{033} were measured at $q = 2.60$ and $4.49$ $\text{Å}^{-1}$, respectively. These reflections are consistent with the d-spacing of the corresponding X-ray diffraction peaks in Fig. 4(e). The diffraction pattern from the $\delta''$ phase, shown in Fig. 5(b), exhibits similar d-spacings to that of the $\delta'$ phase. However, the diffraction pattern for the $\delta''$ phase does not exhibit the same extent of hexagonal-like symmetry that is seen in the $\delta'$ phase. This trend is consistent with the hypothesis that the $\delta''$ phase is lacking the interlayer order of the $\delta'$ phase. Thus, we are able to demonstrate that is possible to not only predict which structures appear during the reaction, but that the metastable phases can in fact be isolated from multistage crystallization and retained.

Overall, in this work, we performed in situ WAXS to monitor the multistage crystallization of manganese oxides under the hydrothermal synthesis condition of $[\text{K}^+] = 0$, 0.2, and 0.33 M. We demonstrated that particle size and off-stoichiometric composition can affect the free-energy landscape of metal oxide precipitation, influencing which metastable phases appear during multistage crystallization, the order in which they appear, and the persistence of these metastable phases. Furthermore, we showed that while a calculated, size-dependent phase diagram qualitatively maps out crystallization pathways under specific reaction conditions, the driving force ($\Delta\Phi$) between two phases can determine the observable progression of crystalline phases by controlling the induction times of various phases. Finally, we successfully isolated the predicted intermediate phases by quenching the reaction at designated time points. Our work provides evidence for the predictive capability of remnant metastability as a general framework not only for predicting synthetically-accessible metastable materials, but also as a practical tool to guide predictive synthesis.

## Methods

### Thermodynamics of the size-dependent Pourbaix grand potential.
The size-dependent Pourbaix grand potential is an extension of the Pourbaix potential[22], but incorporating surface energy contributions, and transformed into a grand potential $\mu_{\text{Bulk}} = \mu_f - \mu_K N_K$ with respect to external K$^+$ activity:

$$\overline{\Phi} = \frac{1}{N_M}\left( (\mu_f - \mu_K N_K - N_O \mu_{\text{H}_2\text{O}}) + (2N_O - N_H)\mu_{\text{H}^+} - (2N_O - N_H + Q)E \right) + \left(\frac{1}{R}\right)\frac{\eta\rho\gamma}{N_M}$$

(1)

The first term is the bulk free energy, and the second term is the surface energy contribution. $\mu_f$ is the molar Gibbs formation energy, $N_{Mn}$, $N_K$, $N_O$, and $N_H$

represent the composition of a K-Mn-O-H phase; and $Q$ is the charge of an aqueous ion. In our phase diagrams, the chemical potential of potassium is $\mu_K = \mu_K^\circ + RT\ln[\text{K}^+]$, where the activity of K$^+$ is approximated by its aqueous concentration ($[\text{K}^+]$). The water chemical potential $\mu_{\text{H}_2\text{O}}$ is referenced using the method reported previously[52]. In the surface energy term, $\gamma$ is the surface energy, $1/R$ represents the specific surface area in units of Area/Volume, $\eta$ is the unitless shape factor of the equilibrium particle morphology, and $\rho$ is the molar volume. Because reaction temperature was not modified between the three pathways, the potential does not include the effect of temperature. We omit pressure, as the pressures in hydrothermal synthesis ($P < 30$ MPa) are too low to influence the relative enthalpies between metal oxides.

### DFT calculation of surface and bulk energies.
DFT calculations were performed using the Vienna Ab-Initio Simulation Package (VASP)[53], using the projector augmented wave (PAW) method with the Strongly Constrained and Appropriately Normed (SCAN)[33] exchange-correlation functional, a reciprocal scape discretization of at least 25 Å$^{-1}$, and convergence to $2 \times 10^{-7}$ eV/atom and 0.02 eV/A on energy and interatomic forces, respectively, following earlier benchmarking on the MnO$_2$ system[23]. Surface energies are computed using surface slabs generated using the efficient creation and convergence scheme developed by Sun and Ceder[54]. K$_x$MnO$_2$-$y$H$_2$O phases are prepared using the techniques described in Ref. [21]. Vibrational entropy contributions to the free-energy are obtained from experimental data, and the formation free-energy is referenced to hydrothermal synthesis conditions (160 °C, see below). Detailed benchmarking of results can be found in Supplementary Note 10 and 11.

### Sample preparation for hydrothermal synthesis.
The reaction conditions on $P_{K=0.2}$ and $P_{K=0.33}$ were adapted from a previously reported procedure[40], in which 1 mmol ($P_{K=0.2}$) or 3 mmol ($P_{K=0.33}$) of KMnO$_4$ (Sigma-Aldrich, 99.0%) was dissolved in 5 mL of 0.2 M HNO$_3$. The purple KMnO$_4$ solution was immediately loaded into a soda lime capillary tube (Kimble 2502, 0.1 mm wall thickness, 75 mm long) which is used as the hydrothermal reactor by syringe. The filling volume of the capillary was approximately 75%. Then, both of the ends of the capillary were flame-sealed. For $P_{K=0}$, the preparation procedure was rather different. In order to create a low-K environment, 0.25 M KMnO$_4$ solution was passed through a column filled with Amberlite® 120 ion-exchange resin in H$^+$ form[55]. As determined by inductively coupled plasma (ICP), [MnO$_4^-$] in the ion-exchanged KMnO$_4$ solution were ~0.08 M, and [K$^+$] is below the detection limit (see Supplementary Table 4 and 5).

Before running in situ reactions, we also ran ex situ hydrothermal syntheses in conventional autoclaves parallel to the in situ reaction conditions. We confirmed the consistency in the end product between the in situ and ex situ experiments. For details of the sample preparation method, see Supplementary Note 7. Additionally, the pH and oxidation reduction potentials (ORP) of the solutions are measured before and after the reaction with a pH/ORP meter and the result is shown in Supplementary Table 1. The pH/ORP values are used as the input parameters in constructing the size-dependent phase diagram and induction time theory.

### In situ X-ray scattering study of reaction pathways.
The in situ wide-angle scattering (WAXS) results were obtained at beamline 11–3 at the Stanford Synchrotron Radiation Lightsource (SSRL), with an incident wavelength of 0.9744 Å. Experimental data were collected by using a Rayonix 225 detector, and calibrated using 325-mesh Si powder. During the hydrothermal reaction, 2D scattering patterns were collected consecutively, with each image exposed for 1 min. X-ray scattering profiles were acquired by integrating individual 2D scattering patterns over polar angles, $30° \leq \chi \leq 150°$. To hold the hydrothermal capillary reactor at the beamline, we have designed a sample holder/heater. Similar designs of the reactor[56] has also been applied to other synchrotron facilities for in situ reaction studies[57]. For detailed measurements set-ups, please see Supplementary Figure 1 in Supplementary Note 3. During the hydrothermal reaction, the temperature was kept at 160 °C for 24 h.

### TEM measurements of quenched products.
To study the microstructure of the intermediate phases, we forced the hydrothermal reaction to stop at designated time points by stopping the reaction, opening the reactor and extracting the contents inside. The solution and precipitates in the capillary were dispersed in DI water and then sonicated for 2 min. To prepare the samples for TEM, drops of the dilute solution were transferred onto silicon-nitride TEM grids (SimPore Inc.). TEM analysis was conducted using an FEI Co. Talos F200X transmission electron microscope operating at an accelerating voltage of 200 keV. Structural characterization was performed by acquiring selected area electron diffraction patterns on an FEI Co. Ceta 16 M pixel CMOS camera at a camera length of 660 mm. A platinum standard was used to calibrate the camera constant, allowing SAED reflections to be accurately measured and indexed.

### Data availability.
All data needed to evaluate the conclusions in the paper, including the cifs of the crystalline structures discussed in this article, are present in the paper and/or the Supplementary Information. Additional data related to this paper are available as requested from the authors.

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

## Acknowledgements

This work was supported by Center for Next Generation Materials by Design: Incorporating Metastability, an Energy Frontier Research Center funded by the U.S. Department of Energy, Office of Science, Basic Energy Sciences under Award No. DE-AC36-08GO28308. Use of the Stanford Synchrotron Radiation Lightsource, SLAC National Accelerator Laboratory, was supported by the U.S. Department of Energy, Office of Basic Energy Sciences under Contract No. DE-AC02-76SF00515. D.K.

acknowledges computational resources provided by the National Energy Research Scientific Computing Center (NERSC), a DOE Office of Science User Facility supported by the Office of Science of the U.S. Department of Energy under Contract No. DE-AC02-05CH11231, as well computational resources sponsored by the Department of Energy's Office of Energy Efficiency and Renewable Energy and located at the National Renewable Energy Laboratory. The authors would like to thank Charles Troxel Jr. and Ross Arthur for helping build the hydrothermal reactor, and Tim Dunn for beamline support.

## Author contributions

W.S., L.S., G.C., M.T., K.S., D.K., L.G., and D.G. originated the research. B.-R.C., L.S., and K.S. performed in situ X-ray diffraction and data analysis. K.S. designed the hydrothermal reactor, then K.S., V.T., and L.S. tested it. W.S., D.K., and G.C. developed the theory and computational tools for the size-dependent phase diagram and crystallization pathways. J.M. and B.G. contributed to the TEM characterization. The manuscript was primarily written by B.-R.C., W.S., L.S., and K.S. All authors contributed to discussions of data and manuscript review.

## Additional information

**Competing interests:** The authors declare no competing interests.

