## [Peer Review File · Nature Communications]

REVIEWERS' COMMENTS:

Reviewer #1

Most of my comments have been addressed in the revised version.

Reviewer #2

In this paper, the authors investigate the formation of different polymorphs of (potassium)-manganese dioxide via hydrothermal synthesis as a function of nanoparticle size and potassium concentration. From theoretical considerations of how the surface energy of a nanoparticle can change the relative ordering of stability of different polymorphs, they predict which polymorphs and phase transitions are possible as growth proceeds. They test the theoretical predictions by experimental synthesis and x-ray crystallography to determine the phase(s) present.

While the basic ideas of this paper are straightforward and clear, this is the first paper in which I have seen all of these ideas synthesized. The ideas show a path toward which the synthesis of particular desired oxide polymorphs might be achieved. The experimental validation is especially noteworthy. For these reasons, my opinion of the paper in its current form is favorable.

In a work of this scope, there are inevitably some loose ends, noted by the earlier referees. It is unfortunate that some polymorphs predicted to be stable at a certain particle size are not observed experimentally, but the authors note the difficulty in fully predicting growth dynamics ab initio. I feel that the authors have addressed all of the criticisms of the earlier referees; for example, the fact that they can not reliably calculate the surface energies of hydrated oxides due to water dissociation is plausible to me as I have observed the same thing happen in my studies.

I remain a bit confused by the proliferation of K-MnO₂ phases discussed in the text, some of which may not have been discussed before in the literature. It would be helpful to extend Table S1 in the Supplementary Information or add a new table there in which all of the discussed phases are listed with basic crystallographic parameters, whether they were previously known or introduced in this paper, and notes about disorder present if any.

Also, Reference 5 in the text has a title repeated, and Ref. 9 of the SI is missing an article number.

Revisions and Responses to the Reviewers' Comments

Manuscript #: NCHEM-17112457A-Z

Title: Understanding Crystallization Pathways Leading to Manganese Oxide Polymorph Formation

Author(s): Bor-Rong Chen^{1,#}, Wenhao Sun^{2,3,#}, Daniil A. Kitchaev⁴, John S. Mangum⁵, Vivek Thampy¹, Lauren M. Garten⁶, David G. Ginley⁶, Brian P. Gorman⁵, Kevin H. Stone¹, Gerbrand Ceder^{2,3,*}, Michael F. Toney^{1,7,*}, Laura T. Schelhas^{7,*}

We thank the referees for reviewing our manuscript. We appreciate the constructive comments from the reviewers on our work and we have revised the manuscript according to their suggestions.

The following is a point-by-point response and revisions to the manuscript:

Reviewer #1

Most of my comments have been addressed in the revised version.

Response: We thank Reviewer #1 for the effort on reviewing the revised version.

Reviewer #2

In this paper, the authors investigate the formation of different polymorphs of (potassium)-manganese dioxide via hydrothermal synthesis as a function of nanoparticle size and potassium concentration.

From theoretical considerations of how the surface energy of a nanoparticle can change the relative ordering of stability of different polymorphs, they predict which polymorphs and phase transitions are possible as growth proceeds. They test the theoretical predictions by experimental synthesis and x-ray crystallography to determine the phase(s) present.

While the basic ideas of this paper are straightforward and clear, this is the first paper in which I have seen all of these ideas synthesized. The ideas show a path toward which the synthesis of particular desired oxide polymorphs might be achieved. The experimental validation is especially noteworthy. For these reasons, my opinion of the paper in its current form is favorable.

In a work of this scope, there are inevitably some loose ends, noted by the earlier referees. It is unfortunate that some polymorphs predicted to be stable at a certain particle size are not observed experimentally, but the authors note the difficulty in fully predicting growth dynamics ab initio. I feel that the authors have addressed all of the criticisms of the earlier referees; for example, the fact that they cannot reliably calculate the surface energies of hydrated oxides due to water dissociation is plausible to me as I have observed the same thing happen in my studies.

I remain a bit confused by the proliferation of K-MnO₂ phases discussed in the text, some of which may not have been discussed before in the literature. It would be helpful to extend Table S1 in the

Supplementary Information or add a new table there in which all of the discussed phases are listed with basic crystallographic parameters, whether they were previously known or introduced in this paper, and notes about disorder present if any.

Also, Reference 5 in the text has a title repeated, and Ref. 9 of the SI is missing an article number.

Response: We thank the reviewer for reviewing and recognizing our work. The discussion of the K-MnO₂ phases (a.k.a δ -family of phases) in the article was confusing for the readers. Therefore, we added Supplementary Table 2 (page 3 of SI) to summarize the crystallographic parameters and brief descriptions of the phase for all of the phases discussed in this work. We hope the addition of this table will make the discussion easier to follow.

We have also addressed the typos in the reference lists, and we thank the reviewer for catching this.